# Rapid differentiation of *Staphylococcus aureus* in blood cultures using the STAPH score: a prospective observational study

**Toshimasa Hayashi,**[1] **Masakazu Yoshida**[2]

**ABSTRACT** Rapid and accurate identification of *Staphylococcus aureus* (SA) in blood culture specimens is crucial for timely clinical intervention. Traditional Gram staining methods, although widely accessible and cost-effective, exhibit variable sensitivities and specificities in the BACTEC system. We aimed to develop and validate the "STAPH score," a novel semi-quantitative scoring system that integrates Gram stain morphology and time to positivity to differentiate SA from coagulase-negative staphylococci (CoNS) in blood cultures. We analyzed 148 blood culture specimens from septic patients. Each specimen was assessed independently by two examiners using Gram staining and the STAPH score, which assigns points based on five parameters: cluster size, time to positivity, aerobic enlargement, pint (three-dimensional appearance), and the presence of hemorrhage. Sensitivity and specificity were calculated for various cutoff points. Cohen's kappa coefficient was used to assess inter-examiner agreement. Of the 148 specimens, 49 (33.1%) were identified as SA and 99 (66.9%) as CoNS. At a STAPH score cutoff of 3 points, the sensitivity was 93.9% (95% confidence interval [CI], 83.1%–98.7%) and specificity was 91.9% (95% CI, 84.7%–96.4%). The kappa coefficient at this cutoff was 0.67 (95% CI, 0.55–0.79). A STAPH score <3 effectively ruled out SA with 100% sensitivity, whereas a score of 5 confirmed SA with 100% specificity. The STAPH score is a reliable and efficient tool for the rapid identification of SA in blood cultures within the BACTEC system. By combining Gram staining observations with time to positivity, this method enhances diagnostic accuracy, reduces subjectivity, and supports timely clinical decision-making.

**IMPORTANCE** The rapid and accurate identification of *Staphylococcus aureus* (SA) in blood cultures is vital for timely and appropriate clinical intervention. This study introduces the "STAPH score," a novel semi-quantitative scoring system that combines Gram stain morphology and time to positivity. By providing a reliable and efficient method to differentiate SA from coagulase-negative staphylococci, the STAPH score enhances diagnostic accuracy and reduces subjectivity in microscopic examinations. This score, applicable within the BACTEC system, addresses the limitations of traditional Gram staining methods and expensive molecular techniques. The implementation of the STAPH score in clinical practice can lead to faster diagnosis, improved patient outcomes, and optimized antimicrobial therapy. This method is particularly valuable in resource-limited settings where advanced diagnostic tools may not be available.

**KEYWORDS** *Staphylococcus aureus* bacteremia, rapid diagnostic testing, Gram staining, time to positivity

The mortality rate of *Staphylococcus aureus* (SA) bacteremia is high, particularly in sepsis cases, where the mortality rate can be 3 to 4 times higher (1, 2). This type of sepsis involves multiple virulence factors possessed by SA. In cases of bacteremia, SA's coagulase, von Willebrand factor-binding protein, and clumping factor protein

Address correspondence to Toshimasa Hayashi, t-hayashi@maebashi.jrc.or.jp.

The authors declare no conflict of interest.

promote fibrin clot formation and bacterial aggregation, creating an environment conducive to bacterial survival in the bloodstream (3). Therefore, determining whether the bacteria identified in blood cultures are SA is crucial for further diagnostic evaluation for disseminated infections and treatment decisions (4). However, identifying SA from positive cultures requires additional time-consuming and costly tests, and coagulase-negative staphylococci can grow as contaminants, mimicking SA. These factors highlight the necessity for reliable and rapid identification methods.

In blood culture diagnostics, the observation of Gram-positive cocci clusters suggestive of *Staphylococcus* species necessitates accurate identification, especially distinguishing SA from coagulase-negative staphylococci (CoNS), owing to their distinct clinical implications. The detection of SA in a single set among multiple blood culture sets typically signals clinically significant bacteremia that requires intervention (5). In contrast, the detection in just one set of CoNS, predominantly considered skin flora, is often considered a result of contamination (6, 7). Rapid and accurate identification of SA directly from blood cultures is essential for timely clinical intervention but poses a significant challenge without the use of expensive equipment or extensive additional testing. Several methods have been investigated for the rapid differentiation of SA, including fluorescent *in situ* hybridization, nucleic acid amplification techniques, and mass spectrometry (8). Although these methods can provide accurate and rapid results, they often require specialized equipment and significant training and can be costly, making them less accessible for routine clinical use. For instance, fluorescent *in situ* hybridization is highly specific but requires specialized equipment and training. Nucleic acid amplification techniques, including PCR, offer rapid results but can be costly and susceptible to contamination. Mass spectrometry, although effective, is limited by its high cost and need for experienced personnel. In contrast, Gram staining is widely accessible, immediate, and cost-effective, which makes it ideal for routine clinical use. Previous studies have reported the possibility of distinguishing SA based on the size of the bacteria, the number of bacteria forming clusters, and the presence of surrounding redness in Gram-stained specimens of positive blood cultures (9, 10). However, traditional Gram staining methods, when applied within the BACTEC system, have previously exhibited unacceptable levels of variability in sensitivity and specificity.

In this study, we aimed to address these limitations by developing a standardized, quantifiable scoring system, the "STAPH score," to enhance the accuracy of SA detection using the BACTEC system, thereby significantly improving clinical diagnostics. The STAPH score assigns points based on five parameters observed in Gram stain microscopy of blood culture specimens and culture positivity time: size of clusters, time to positivity, aerobic enlargement, pint (three-dimensional appearance), and the presence of hemorrhage.

## MATERIALS AND METHODS

### Sample collection and culture conditions

From 1 July 2023 to 31 December 2023, we collected 180 blood culture specimens from patients with sepsis or suspected sepsis at our institution. The exclusion criteria were as follows:

1. Specimens showing concurrent growth of SA and CoNS from the same bottle (0 specimens).
2. Specimens with the growth of non-*Staphylococcus* species from the same bottle (4 specimens).
3. Specimens with insufficient bacterial quantity to confirm the presence of bacteria under 10 fields of view (5 specimens).
4. Specimens collected confirming negativity within 14 days after a previous positive *Staphylococcus* culture (22 specimens were excluded due to potential alterations

in bacterial morphology or growth times from antibiotic treatment, which are unsuitable for a pure study of the STAPH score. Clinically, previous results can be referenced within 2 weeks).

5. Specimens submitted in pediatric blood culture bottles (1 specimen).

After applying these criteria, 148 specimens remained for analysis. Blood cultures were processed using a BD BACTEC FX system (Becton Dickinson, Japan) equipped with resin-containing non-hemolytic bottles (aerobic: BD BACTEC 23F bottles; anaerobic: BD BACTEC 22F bottles). The cultures were incubated for up to 7 days, and the time to positivity was recorded. For the STAPH score, we adopted the time to positivity for each sample individually. Species identification was performed using a MALDI Biotyper mass spectrometry system (Becton Dickinson, Japan). All patient information was anonymized, and each specimen was assigned a test date and number.

## Microscopic examination

Gram staining and microscopic examinations were performed according to standardized methods. For each specimen, 10 µL of the sample was smeared onto a glass slide and stained using the Hucker method. Without prior knowledge of the bacterial species, two examiners independently examined the slides at a magnification of 1000×, observing 20–40 fields per specimen. Examiner 1 (TH) is a physician in the Department of Infectious Diseases with 9 years of experience in Gram staining, including routine microscopic interpretation of blood culture specimens. Examiner 2 (MY) is a clinical laboratory technician in the Department of Microbiology with 14 years of experience, routinely performing Gram staining microscopy. After their examination, two novice physicians, one in Postgraduate Year 1 (PGY-1) and one in Postgraduate Year 2 (PGY-2), used the STAPH score to evaluate the last 40 samples collected during the study period. This was performed to assess how effectively novice physicians could use the STAPH score to identify SA.

## STAPH score assessment

Each specimen was evaluated using the STAPH score, which ranges from 0 to 5 points based on specific criteria. Each parameter is defined as follows:

1. Size of clusters: Average number of bacteria in clusters over 10 fields >16 (+1 point).
2. Time to positivity: Culture becomes positive in <16 h (+1 point).
3. Aerobic enlargement: Diameter of the largest bacterium in aerobic bottles ≥1 µm or diameter of the smallest bacterium in anaerobic bottles <1 µm (+1 point).
4. Pint: Clusters appear three-dimensional (+1 point).
5. Hemorrhage: Pink-colored coagulation clots or membranes present around the bacteria (+1 point).

These criteria are listed in Table 1. Each examiner independently assessed the specimens according to these criteria. Examples of Gram staining images and schematic diagrams illustrating these criteria are shown in Fig. 1.

## Statistical analysis and ethics approval

We compared STAPH scores between the SA and CoNS groups. The sensitivity and specificity of each threshold (ranging from 0 to 5 points) were initially calculated based on the individual scores of the two examiners. The sensitivity and specificity of each examiner were used to perform separate receiver operating characteristic (ROC) analyses to evaluate the diagnostic performance. We then recalculated the sensitivity and specificity using the average STAPH scores from both examiners to determine the optimal threshold using the Youden index. Cohen's kappa coefficient was calculated

**TABLE 1** : STAPH score criteria

| Parameter | Condition | Points |
|---|---|---|
| S: Size of clusters | Average number of bacteria in clusters over 10 fields >16 | +1 |
| T: Time to positivity | Culture becomes positive in <16 h | +1 |
| A: Aerobic enlargement | Diameter of the largest bacterium in aerobic bottles ≥1 µm or diameter of the smallest bacterium in anaerobic bottles <1 µm | +1 |
| P: Pint | Clusters appear three-dimensional | +1 |
| H: Hemorrhage | Pink-colored coagulation clots or membranes present around the bacteria | +1 |

to assess the inter-examiner agreement at this threshold. A multivariate analysis was performed to evaluate the influence of each STAPH score parameter on SA identification, and each score was treated as an independent variable. A significance level of $P < 0.05$ was used for all statistical tests. The statistical analyses were performed using R (version 4.3.3) with EZR (version 1.64) (11). This study was reviewed by the hospital's medical ethics committee on 4 January 2024 and approved on 25 January 2024 (Approval number 2023-57). An amendment to conduct additional research was submitted on 30 June 2024 and was approved on 4 July 2024. The need for patient consent was waived by the ethics committee.

## RESULTS

A total of 148 specimens were analyzed and collected from 68 patients. Of these, 49 were identified as SA (33.1%) and 99 as CoNS (66.9%). One patient was a repeat case, with a second sample collected 37 days after the initial positive result, which grew a different organism. The median time to positivity was 13.0 h (range: 6.9–84.6 h) for the SA group and 20.9 h (range: 2.3–76.7 h) for the CoNS group. Among the specimens with a growth time of 14 to <16 h, 42.9% were identified as SA, whereas in the time range of 16 to <18 h, the proportion of SA dropped to 15.8%. A histogram of up to 18 h of the time to positivity is shown in Fig. 2.

For SA, the two examiners recorded average STAPH scores of 4.3 and 4.2, with median values of 5.0 and 4.0, respectively. Similarly, for CoNS, the average scores were 1.2 and 1.6, with median values of 1.0 and 2.0.

The sensitivity and specificity at various cutoff values, as well as the combined diagnostic performance based on the average STAPH scores from both examiners at the final selected cutoff value, are summarized in Table 2. The ROC analysis results showed that the area under the curve was 0.964 (95% confidence interval [CI], 0.941–0.988) for Examiner 1 and 0.971 (95% CI, 0.948–0.994) for Examiner 2. Using the examiners' combined average value, when the STAPH score cutoff was set at 3 points, the sensitivity was 93.9% (95% CI, 83.1%–98.7%), and the specificity was 91.9% (95% CI, 84.7%–96.4%), enabling the distinction of SA with the highest Youden index of 0.86. We calculated the required sample size to achieve these sensitivity and specificity values, which was 43 for sensitivity and 88 for specificity (12). Therefore, we determined that the sample size of 148 was sufficient for this study. At a cutoff of 2 points, the sensitivity was 100% (95% CI, 89.4%–100%) and the specificity was 68.7% (95% CI, 58.6%–77.6%), while a cutoff of 4 points yielded a sensitivity of 75.5% (95% CI, 61.1%–86.7%) and a specificity of 100% (95% CI, 94.6%–100%). When the cutoff value was set at 3, the kappa coefficient was 0.67 (95% CI, 0.55–0.79), indicating "substantial" agreement between the examiners (8).

At a STAPH score cutoff of 3 points, the PGY-1 physician's results showed a sensitivity of 100% (95% CI, 66.1%–100%) and a specificity of 59.3% (95% CI, 38.8%–77.6%). The PGY-2 physician's results showed a sensitivity of 100% (95% CI, 66.1%–100%) and a specificity of 63.0% (95% CI, 42.4%–80.6%). Their results were identical to those of the authors in that a score of 0–2 points yielded 100% sensitivity and a score of 5 points ensured 100% specificity.

The results of the multivariate analysis of the STAPH score parameters, including odds ratios, CIs, and $P$-values for each STAPH score component, are presented in Table 3.

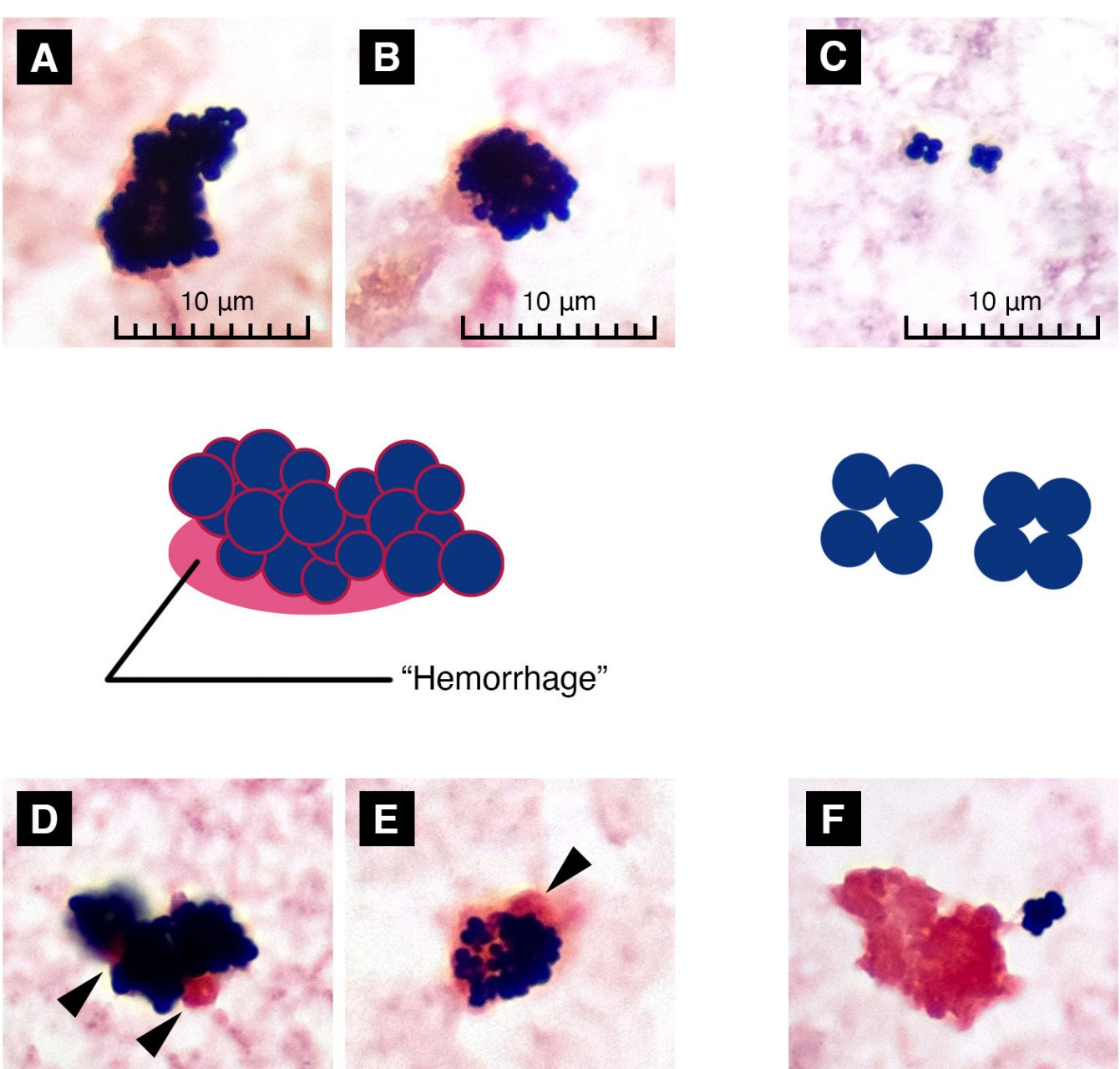

**FIG 1** Examples of Gram staining images and schematic diagrams illustrating the STAPH score criteria. (A) Gram stain image of *Staphylococcus aureus* (SA) grown in an aerobic bottle, showing large, three-dimensional clusters that are difficult to bring into focus. (B) Gram stain image of SA grown in an anaerobic bottle, showing smaller bacteria. (C) Gram stain image of coagulase-negative staphylococci (CoNS) (*S. epidermidis*) grown in an aerobic bottle, with less dense clusters. (D and E) Gram stain image showing the presence of hemorrhage, appearing as if pink-colored coagulation clots are adjacent to or pink-colored membranes are present around SA clusters (arrowheads). (F) Coagulation clots are not directly attached to the bacteria, which are judged not to be indicative of hemorrhage (*S. epidermidis*). All images were observed at 1,000 × magnification. The schematic diagrams in the middle illustrate the criteria used in the STAPH score assessment: size of clusters, (time to positivity,) aerobic enlargement, pint (three-dimensional appearance), and presence of hemorrhage.

This table illustrates the statistical significance and strength of the association between each score component and the SA identification outcome. These results highlight the importance of "Time to positivity," "Pint," and "Hemorrhage" for Examiner 1 and "Aerobic enlargement" and "Hemorrhage" for Examiner 2 in the accurate identification of SA in blood cultures.

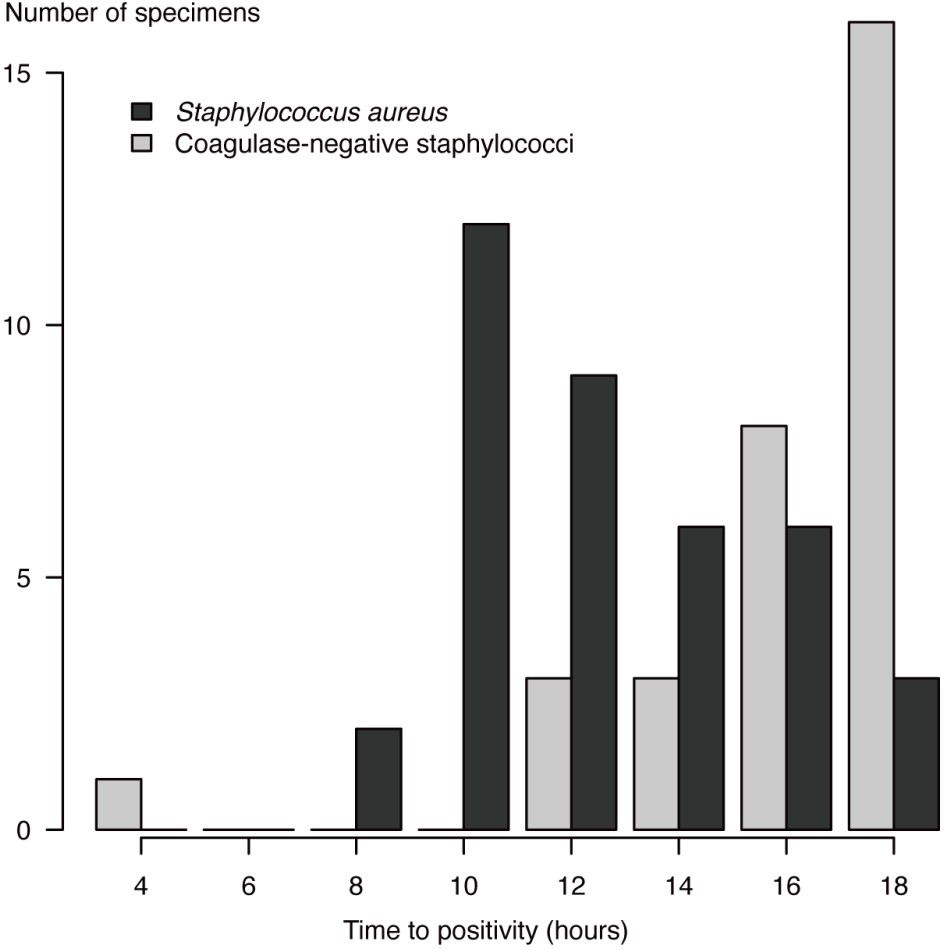

**FIG 2** Distribution of the time to positivity for *Staphylococcus aureus* (SA) and coagulase-negative staphylococci (CoNS) in blood cultures within 18 h. Histogram showing the time to positivity for 148 blood culture specimens within the first 18 h. SA is represented in black, and CoNS are represented in gray. The proportion of SA was the highest (42.9%) from 14 to <16 h but significantly decreased to 15.8% between 16 and 18 h. The *X*-axis shows the time to positivity in hours, and the *Y*-axis shows the number of specimens.

## DISCUSSION

This study yielded two significant findings. First, we demonstrated that rapid and accurate differentiation of SA is feasible within the BACTEC system by integrating several common microscopic observations with culture positivity times. This approach enables timely and appropriate clinical intervention, as the STAPH score can be calculated as part of routine testing procedures without requiring additional rapid testing methods. Second, the STAPH score, which was developed based on straightforward and memorable criteria, effectively reduces the subjectivity associated with microscopic examinations and enhances the reproducibility of test results among examiners.

The inclusion of culture positivity time as a criterion is particularly notable, as it compensates for the variability in Gram staining characteristics observed between BACTEC and other systems, such as BacT/ALERT. This integration addresses previous limitations and offers a robust method for accurate SA identification. Furthermore, our multivariate analysis highlights the significance of each STAPH score parameter, underscoring the importance of criteria such as "Hemorrhage" and "Pint" in differentiating SA from CoNS. Our findings suggest that the STAPH score is a valuable tool in clinical settings to facilitate rapid decision-making regarding antimicrobial therapies. By

**TABLE 2** : Sensitivity and specificity of the STAPH score at different cutoff values[a]

| Cutoff value | Examiner 1 (TH) | | Examiner 2 (MY) | | Examiners' combined average | |
|---|---|---|---|---|---|---|
| | Sensitivity (95% CI) (%) | Specificity (95% CI) (%) | Sensitivity (95% CI) (%) | Specificity (95% CI) (%) | Sensitivity (95% CI) (%) | Specificity (95% CI) (%) |
| 0 | 100 (89.4–100) | 0 (0.0–5.4) | 100 (89.4–100) | 0 (0.0–5.4) | 100 (89.4–100) | 0 (0.0–5.4) |
| 1 | 100 (89.4–100) | 33.3 (24.2–43.5) | 100 (89.4–100) | 14.1 (8.0–22.6) | 100 (89.4–100) | 28.3 (19.7–38.2) |
| 2 | 100 (89.4–100) | 64.6 (54.4–74.0) | 100 (89.4–100) | 47.5 (37.3–57.8) | 100 (89.4–100) | 68.7 (58.6–77.6) |
| 3 | 95.9 (86.0–99.5) | 84.8 (76.2–91.3) | 95.9 (86.0–99.5) | 82.8 (73.9–89.7) | 93.9 (83.1–98.7) | 91.9 (84.7–96.4) |
| 4 | 77.6 (63.4–88.2) | 93.9 (87.3–97.7) | 81.6 (68.0–91.2) | 99.0 (94.5–100) | 75.5 (61.1–86.7) | 100 (94.6–100) |
| 5 | 57.1 (42.2–71.2) | 100 (94.6–100) | 42.9 (28.8–57.8) | 100 (94.6–100) | 34.7 (21.7–49.6) | 100 (94.6–100) |

[a]Note: The examiners' combined average shows the sensitivity and specificity, calculated by averaging the STAPH scores of examiners 1 and 2 for each specimen (listed only if the average is an integer).

prioritizing the sensitivity of our scoring system, we ensured that potential SA infections were not missed, thereby improving patient outcomes.

Gram staining of SA in blood cultures using the BACTEC system typically showed larger bacteria in aerobic bottles and smaller bacteria in anaerobic bottles, with clusters appearing three-dimensional enough to challenge focus adjustment and surrounded by red-stained fibrin clots. Compared to previous studies using the BacT/ALERT system (10), differences in cluster and bacterial sizes between SA and CoNS were minimal in the BACTEC system, suggesting that these observations alone might not provide sufficient accuracy for distinguishing SA. Therefore, the positivity time was introduced as an additional criterion. Although previous studies have reported a high likelihood of SA when the positivity time was <18 h (13, 14), this study found that the probability of SA dropped to 33.1% when the positivity time exceeded 16 h, leading us to set the positivity time criterion to <16 h. When we changed the criterion from <16 h to <18 h and examined the sensitivity and specificity of the STAPH score, the specificity decreased from 91.9% to 89.9%, while the sensitivity remained at 93.9%. Additionally, although it seemed that the probability of false negatives would increase when SA grew beyond 16 h, the number of false negatives was minimized using other scores based on Gram stain observations. Specifically, by integrating these additional criteria, we reduced the number of false negatives from 14 to only 3 out of 49 SA samples at the cutoff of 3 points. To further enhance the predictive accuracy, based on literature reporting on the "oozing sign" associated with SA (9), we added "Hemorrhage" as a criterion. The multivariate analysis of the STAPH score parameters revealed that "Size of clusters" did not demonstrate significant differences between the two examiners, underscoring the challenges of distinguishing SA within the BACTEC system. However, the presence of this parameter suggests that the STAPH score could potentially be effective even in the BacT/ALERT system. The "Hemorrhage" factor, found significant by both examiners, has been demonstrated to be effective in differentiating SA not only in the BacT/ALERT system but also in the BACTEC system.

These innovations demonstrate that the STAPH score has equal or superior accuracy compared to existing evaluation methods. For example, the study by Hadano et al. (9) evaluating the Oozing sign reported a sensitivity of 78.7% and specificity of 95.0%, and calculations based on the study by Zimerman et al. (13) evaluating time to positivity

**TABLE 3** : Results of multivariate analysis of the STAPH score parameters

| Parameter | Examiner 1 (TH) | | Examiner 2 (MY) | |
|---|---|---|---|---|
| | Odds ratio (95% CI) | P-value | Odds ratio (95% CI) | P-value |
| S: Size of clusters | 1.77 (0.331–9.49) | 0.504 | 3.3 (0.252–43.1) | 0.363 |
| T: Time to positivity | 11.4 (2.57–51.0) | <0.05 | 6.52 (0.933–45.5) | 0.0587 |
| A: Aerobic enlargement | 1.89 (0.330–10.9) | 0.474 | 13 (1.65–102) | <0.05 |
| P: Pint | 24.6 (3.94–153) | <0.05 | 4.01 (0.65–24.7) | 0.135 |
| H: Hemorrhage | 9.67 (2.55–36.7) | <0.05 | 286 (27.4–2970) | <0.05 |

showed a sensitivity of 65.7% and specificity of 90.4%. In contrast, the STAPH score showed a sensitivity of 93.9% and specificity of 91.9% at a cutoff of 3 points. Additionally, as a quantitative test, it demonstrated 100% sensitivity at 0–2 points and 100% specificity at 5 points, indicating its potential for higher accuracy. Considering the clinical significance of SA, it was essential to prioritize sensitivity; hence, the cutoff value was set at ≥3 points. At this cutoff, the agreement between examiners was "substantial," validating the universal applicability of the STAPH score. The evaluation by novice physicians yielded similar results to those of the authors at 0–2 points and 5 points. While the sample size is limited, these findings suggest that the STAPH score functions effectively regardless of the level of experience.

Additionally, re-examination of specimens scored between 3 and 4 points that were not SA revealed that clusters of *S. hominis* and *S. epidermidis* could be large and three-dimensional and easily mistaken for SA in aerobic bottles because of their size, whereas clusters of *S. capitis*, although flat, were large and had small bacterial sizes, making them easy to mistake for SA in anaerobic bottles.

One of the important points in evaluating the STAPH score was the subjectivity in evaluating aerobic enlargement. We have been using a calibrated electronically displayed scale bar with 1 µm increments to assess the presence of aerobic enlargement. This objective size criterion allows us to make accurate measurements without waiting for both the aerobic and anaerobic bottles in the same set to show bacterial growth. However, we recognize that if the scale bar cannot be utilized, it would necessitate waiting for both bottles to become positive to compare the sizes of the bacteria, which could delay the scoring process and impact the rapidity and accuracy of the system. To mitigate this potential drawback, we propose an alternative approach. We suggest preparing "standard" slides from both the aerobic and anaerobic bottles where SA was previously detected in blood cultures. These standard slides can be used as controls for size comparison, ensuring that the evaluation of aerobic enlargement remains objective and consistent even in the absence of a scale bar. This approach will help maintain the reliability and efficiency of the STAPH scoring system in various clinical settings.

A major limitation of this study is that the STAPH score has been validated only within our institution, so its clinical application may be restricted until its reliability is confirmed through further studies at other facilities and by more examiners using these criteria in their daily workflow. Additionally, the inherently subjective nature of the interpretation of Gram staining remains a challenge. However, averaging scores from multiple examiners could reduce subjectivity, as demonstrated in this study, where the average scores of two examiners provided higher diagnostic accuracy than scores from a single examiner. Recent advances in artificial intelligence (AI) research on bacterial identification suggest that integrating the STAPH score into AI assessments could lead to more objective and accurate bacterial identification in the future. Moreover, there is also the limitation of the absence of an investigation into the use of hemolytic bottles and pediatric blood culture bottles. Further studies should explore the applicability of the STAPH score across different clinical settings and blood culture systems to validate and refine its effectiveness.

## Conclusion

The STAPH score developed in this study represents a straightforward approach to differentiate SA from blood culture specimens within a routine diagnostic workflow. This method stands out for its rapid turnaround time and does not require specialized techniques or any special equipment beyond what is typically used in a standard microbiology laboratory, such as blood culture devices, aerobic and anaerobic blood culture bottles, a Gram staining kit, and a microscope capable of 1,000 × magnification, making it highly practical in clinical settings. By integrating Gram staining observations with the time to positivity, our scoring system achieves a superior balance of sensitivity and specificity compared to existing methods and demonstrates "substantial" inter-examiner agreement. These findings highlight the potential of the STAPH

score as a reliable and efficient tool for the rapid identification of SA, contributing to timely and appropriate clinical interventions. By implementing this score, clinicians can make informed decisions regarding antimicrobial therapy at an early stage, potentially influencing clinical practice by promoting the appropriate use of antimicrobials and shaping public policies for the management of infectious diseases.

## ACKNOWLEDGMENTS

We would like to thank Dr. Mariko Yamada (PGY-1) and Dr. Hayato Shioya (PGY-2) for their cooperation and contributions to this study, particularly in the evaluation of the STAPH score with the last 40 samples collected during the study period.

This research received no specific grants from any funding agency in the public, commercial, or not-for-profit sectors.

This study and manuscript preparation were led by T.H., who was responsible for the conceptualization, methodology design, data collection, analysis, and writing of the original draft. M.Y. contributed significantly to the research process, including data collection and analysis, and participated in the methodology design.

## AUTHOR AFFILIATIONS

[1]Department of Infectious Diseases, Japanese Red Cross Maebashi Hospital, Maebashi, Japan
[2]Department of Microbiology, Japanese Red Cross Maebashi Hospital, Maebashi, Japan

## AUTHOR ORCIDs

Toshimasa Hayashi http://orcid.org/0000-0001-8803-0914

## AUTHOR CONTRIBUTIONS

Toshimasa Hayashi, Conceptualization, Data curation, Formal analysis, Methodology, Project administration, Supervision, Writing – original draft, Writing – review and editing | Masakazu Yoshida, Data curation, Methodology, Resources

## ADDITIONAL FILES

The following material is available online.

Open Peer Review

**PEER REVIEW HISTORY (review-history.pdf).** An accounting of the reviewer comments and feedback.

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
