## [Reviewer comments · Microbiology Spectrum]

Microbiology Spectrum

Rapid differentiation of *Staphylococcus aureus* in blood cultures using the STAPH score: a prospective observational study

Toshimasa Hayashi and Masakazu Yoshida

Corresponding Author(s): Toshimasa Hayashi, Japanese Red Cross Maebashi Hospital

Review Timeline:

Submission Date:	May 23, 2024
Editorial Decision:	June 26, 2024
Revision Received:	July 9, 2024
Accepted:	July 25, 2024

Editor: Deena Altman

Reviewer(s): Disclosure of reviewer identity is with reference to reviewer comments included in decision letter(s). The following individuals involved in review of your submission have agreed to reveal their identity: Jia jia Yin (Reviewer #1); Bryan H Schmitt (Reviewer #2)

Transaction Report:

DOI: <https://doi.org/10.1128/spectrum.01223-24>

Re: Spectrum01223-24 (Rapid differentiation of *Staphylococcus aureus* in blood cultures using the STAPH score: a prospective observational study)

Dear Dr. Toshimasa Hayashi:

Thank you for the privilege of reviewing your work. Below you will find my comments, instructions from the Spectrum editorial office, and the reviewer comments.

Revision Guidelines

Sincerely,
Deena Altman
Editor
Microbiology Spectrum

Reviewer #1 (Comments for the Author):

The authors designed STAPH score, a non-technical, affordable tool suitable for clinical settings. Its objective criteria lessen subjectivity, demonstrating high specificity and sensitivity, making it a reliable tool for identifying SA swiftly, facilitating prompt clinical intervention. With this score, clinicians can promptly determine antimicrobial therapy, demonstrating the potential of STAPH score as an effective rapid identification tool for SA. However, I have the following queries:

- 1.The correlation between Staphylococcus aureus and sepsis should be elucidated in the introduction. Elucidate blood culture identification of negative and positive outcomes, its benefits and constraints.
- 2.The majority of patients in the 180 samples exhibited sepsis or suspected sepsis. Is the sample size sufficient?
- 3.The quantifiable scoring system indicated a positive neutral time range of 14~16h for SA group with mild infection symptoms and low Staphylococcus aureus activity. Will the extended time to a positive result yield false negatives?
- 4.The author stated that the scoring system does not necessitate additional equipment or reagents, yet aerobic and anaerobic culture consumables and microscopes are utilized. Please clarify.
- 5.Since the established scoring system has not been implemented in other institutions, is its application restricted and the result more precise than the existing evaluation method?
- 6.Does the scoring mechanism function for novice physicians?
- 7.I recommend a thorough review of the text, possibly by a native English speaker.

Reviewer #2 (Comments for the Author):

The manuscript details a method to differentiate Staphylococcus aureus from coagulase negative staphylococci growing in blood cultures by using a point based system based on Gram stain microscopic examination.

Major comments

1. Of the STAPH criteria, "hemorrhage" is perhaps the most difficult to explain by text alone. The author's should consider providing a representative image to further illustrate this criteria.
2. The criteria of aerobic enlargement is highly subjective as described in the manuscript and additionally would require a positive anaerobic bottle for comparison, which may delay scoring. Have the authors considered an objective measurement, for example width of bacteria in microns, to satisfy the size criteria? Please discuss if waiting for anaerobic bottle positivity is a potential short coming or not further in the manuscript.
3. Lines 130 to 131. Is the median time to positivity for the 1st bottle only? Please clarify.
4. A major limitation of the study is that the data was collected by the authors alone, only one of which likely represents a staff member that would apply these criteria on a routine basis as a clinical laboratory technician. The data and conclusions would overall be strengthened by increasing the number examiner's who would use these criteria during their daily workflow. The authors should consider the inclusions of additional examiner data int he manuscript or alternatively highlighting this as a major limitation.

Minor comments

1. It would be helpful to the reader to establish early on within the main manuscript itself what each of the criteria in "STAPH" stands for rather than stating only in the abstract and tables.
2. Line 105 states that author TH has 9 years of experience in Gram staining. Please also state whether this experience includes microscopic interpretation.
3. Were any repeat patients included in the study? If so, please identify the number or state that no repeat patients were included.

The authors designed STAPH score, a non-technical, affordable tool suitable for clinical settings. Its objective criteria lessen subjectivity, demonstrating high specificity and sensitivity, making it a reliable tool for identifying SA swiftly, facilitating prompt clinical intervention. With this score, clinicians can promptly determine antimicrobial therapy, demonstrating the potential of STAPH score as an effective rapid identification tool for SA. However, I have the following queries:

1. The correlation between Staphylococcus aureus and sepsis should be elucidated in the introduction. Elucidate blood culture identification of negative and positive outcomes, its benefits and constraints.
2. The majority of patients in the 180 samples exhibited sepsis or suspected sepsis. Is the sample size sufficient?
3. The quantifiable scoring system indicated a positive neutral time range of 14~16h for SA group with mild infection symptoms and low Staphylococcus aureus activity. Will the extended time to a positive result yield false negatives?
4. The author stated that the scoring system does not necessitate additional equipment or reagents, yet aerobic and anaerobic culture consumables and microscopes are utilized. Please clarify.
5. Since the established scoring system has not been implemented in other institutions, is its application restricted and the result more precise than the existing evaluation method?
6. Does the scoring mechanism function for novice physicians?
7. I recommend a thorough review of the text, possibly by a native English speaker.

We have revised manuscript #Spectrum01223-24 in line with the reviewers' comments, the specific points raised by the reviewers were written in **RED letters**, and the other changes and corrections in **BLUE letters**. Our responses to the reviewers' comments are as follows (the reviewers' original comments are written in italics):

Reviewer #1-comment 1

The correlation between Staphylococcus aureus and sepsis should be elucidated in the introduction. Elucidate blood culture identification of negative and positive outcomes, its benefits and constraints.

Response: Thank you for your valuable comment. We have revised the introduction to include a detailed explanation of the correlation between *Staphylococcus aureus* (SA) and sepsis. Additionally, we have elaborated on the identification process of blood culture outcomes, discussing both negative and positive results, along with their respective benefits and constraints. We expect these changes to clarify the study's objectives and the necessity for rapid and accurate identification of SA using blood culture specimens. Specifically, we have added the following text at the beginning of the Introduction (see lines 51–60).

Reviewer #1-comment 2

The majority of patients in the 180 samples exhibited sepsis or suspected sepsis. Is the sample size sufficient?

Response: We appreciate your concern regarding the sample size. Among the 180 samples, the largest exclusion group consisted of specimens collected for the purpose of confirming negativity within 14 days after a previous positive *Staphylococcus* culture, accounting for 22 samples. We excluded these specimens because cases that test positive again within 14 days after a previous positive result are likely to show altered bacterial morphology or extended growth times due to the influence of antibiotic treatment. Such cases are not ideal for a pure study of the STAPH score. Additionally, in clinical practice, blood culture growth examples within two weeks can immediately reference the previously detected bacteria. To clarify this, we have added the following to the METHODS section (see lines 100–102).

Furthermore, based on our study results, at a STAPH score cutoff of 3 points, the sensitivity was 93.9% (95% confidence interval [CI], 83.1–98.7%), and specificity was 91.9% (95% CI, 84.7–96.4%). Using Wilson's method for confidence intervals and the statistical software R, we calculated the required sample size to achieve this sensitivity and specificity, which was 43 and 88, respectively. Therefore, we believe that the 148 samples included in our analysis are sufficient for this study. To clarify this, we have added the following to the RESULTS section (see lines 171–173).

Reviewer #1-comment 3

The quantifiable scoring system indicated a positive neutral time range of 14~16h for SA group with mild infection symptoms and low Staphylococcus aureus activity. Will the extended time to a positive result yield false negatives?

Response: Thank you for your insightful comment. We acknowledge that extending the time to a positive result could potentially lead to false negatives, as the score decreases with longer culture positivity times. However, by combining Gram stain observations with culture positivity time in the scoring system, we were able to minimize false negatives. Specifically, out of 49 SA samples, 14 had a culture positivity time of 16 hours or more. By integrating other criteria from the Gram stain observations, we reduced the number of false negatives to only 3 samples. Furthermore, when we extended the cutoff time for the score from <16 h to <18 h, the sensitivity remained unchanged at 93.9%, and the number of false negatives did not increase. However, the specificity decreased from 91.9% to 89.9%. This point has been clarified in the DISCUSSION section (see lines 218–224).

Reviewer #1-comment 4

The author stated that the scoring system does not necessitate additional equipment or reagents, yet aerobic and anaerobic culture consumables and microscopes are utilized. Please clarify.

Response: Thank you for your comment. We appreciate the opportunity to clarify our statement regarding the equipment and reagents needed for the STAPH score. We agree that while aerobic and anaerobic blood culture bottles, as well as microscopes, are required, these are standard tools already available in clinical laboratories. To address this, we have revised our statement to acknowledge the need for these standard tools explicitly. Specifically, the scoring system relies on standard blood culture bottles, a microscope for Gram staining, a Gram staining kit, and a standard incubator for culturing the bottles. Therefore, we believe that the STAPH score can be implemented without requiring additional specialized equipment or reagents beyond what is routinely used in these settings. To clarify this, we have modified the relevant part of the CONCLUSION section (see lines 280–284).

Reviewer #1-comment 5

Since the established scoring system has not been implemented in other institutions, is its application restricted and the result more precise than the existing evaluation method?

Response: We greatly appreciate your insightful comment regarding the broader applicability and comparative accuracy of the STAPH score. To our knowledge, no other scoring system like the STAPH score for differentiating SA in blood culture specimens has been reported. Currently, the STAPH score has been validated only within our institution, so its clinical application may be restricted until its reliability is confirmed through further studies. Nevertheless, the STAPH score shows equal or superior accuracy compared to existing evaluation methods. For example, the study by Hadano et al. evaluating

the Oozing sign reported a sensitivity of 78.7% and specificity of 95.0%, and calculations based on the study by Zimmerman et al. evaluating time to positivity showed a sensitivity of 65.7% and specificity of 90.4%. Therefore, we hope that other institutions will conduct experimental and research-based validation studies based on our findings. We have added these points to the DISCUSSION section to clarify these aspects (see lines 233–240, 264–267).

Reviewer #1-comment 6

Does the scoring mechanism function for novice physicians?

Response:

Thank you for your thoughtful question. We recognize the importance of ensuring that the STAPH score can be effectively used by novice physicians. To address this, we obtained additional approval from the ethics committee to amend our study protocol and conducted a small-scale study. In this study, two novice physicians, one in PGY-1 (Postgraduate Year 1) and one in PGY-2 (Postgraduate Year 2), evaluated the last 40 samples collected during the study period (including 17 SA samples and 23 CoNS samples) using the STAPH score. At a STAPH score cutoff of 3 points, the PGY-1 physician's results showed a sensitivity of 100% (95% confidence interval [CI], 66.1–100%) and a specificity of 59.3% (95% CI, 38.8–77.6%). The PGY-2 physician's results showed a sensitivity of 100% (95% CI, 66.1–100%) and a specificity of 63.0% (95% CI, 42.4–80.6%). Their results were identical to those of the authors in that a score of 0–2 points yielded 100% sensitivity, and a score of 5 points ensured 100% specificity. While these results should be interpreted with caution due to the limited sample size, they suggest that the STAPH score may be highly sensitive and confirm SA at a score of 5 even for novice physicians. This information has been added to the METHODS, RESULTS, and DISCUSSION sections (see lines 120–123, 179–183, 242–245).

Reviewer #1-comment 7

I recommend a thorough review of the text, possibly by a native English speaker.

Response: Thank you for your suggestion. For this revision, we have utilized a professional English proofreading service to review and edit the manuscript thoroughly. We have attached the proofreading certificate for your reference.

Reviewer #2-Major comment 1

Of the STAPH criteria, "hemorrhage" is perhaps the most difficult to explain by text alone. The authors should consider providing a representative image to further illustrate this criteria.

Response: Thank you for your insightful comment. We agree that the "hemorrhage" criterion can be

challenging to explain through text alone. To address this, we have revised Figure 1 to include representative images and schematic diagrams that clearly illustrate the presence and absence of hemorrhage. The revised figure includes a clear label and detailed captions to enhance understanding. The updated figure is included in the revised manuscript (**FIG 1**).

Reviewer #2-Major comment 2

The criteria of aerobic enlargement is highly subjective as described in the manuscript and additionally would require a positive anaerobic bottle for comparison, which may delay scoring. Have the authors considered an objective measurement, for example width of bacteria in microns, to satisfy the size criteria? Please discuss if waiting for anaerobic bottle positivity is a potential short coming or not further in the manuscript.

Response: Thank you for bringing up such an important point. We evaluated the presence of aerobic enlargement by observing a calibrated scale bar displayed electronically in 1-micrometer increments. Without such an objective size criterion, it would be necessary to wait until bacteria are detected in both the aerobic and anaerobic bottles from the same set to compare their sizes, which could potentially delay scoring and affect the rapidity and accuracy of the scoring system. In cases where the scale bar cannot be utilized, an alternative approach could be to create “standard” slides from both aerobic and anaerobic bottles previously found to contain SA in blood cultures and use them as controls for size comparison. We have added this discussion to the DISCUSSION section (see lines 251–263).

Reviewer #2-Major comment 3

Lines 130 to 131. Is the median time to positivity for the 1st bottle only? Please clarify.

Response: Thank you for your question. The two referenced studies used the time to positivity of the first positive bottle for each patient in their analyses. However, we were concerned that knowing whether another sample submitted on the same day might be from the same patient could slightly influence the scoring. Therefore, we adopted the time to positivity for each sample individually as the criterion for the STAPH score. Consequently, the median time to positivity was calculated based on the positivity times of all 148 samples. We have added the following clarification to the relevant part of the METHODS section as follows:

“For the STAPH score, we adopted the time to positivity for each sample individually.”

Reviewer #2-Major comment 4

A major limitation of the study is that the data was collected by the authors alone, only one of which likely represents a staff member that would apply these criteria on a routine basis as a clinical laboratory technician. The data and conclusions would overall be strengthened by increasing the number examiner's

who would use these criteria during their daily workflow. The authors should consider the inclusions of additional examiner data in the manuscript or alternatively highlighting this as a major limitation.

Response:

We completely agree with the reviewer that demonstrating the applicability of the STAPH score by multiple examiners in their daily workflow is desirable. However, due to resource constraints, we cannot verify this with other clinical laboratory technicians within our institution. Therefore, we have emphasized this as a major limitation in the DISCUSSION section (see lines 264–267) and have added a small-scale study to assess the effectiveness of the STAPH score with novice physicians in PGY-1 and PGY-2 (see lines 120–123, 179–183, 242–245).

Reviewer #2-minor comment 1

It would be helpful to the reader to establish early on within the main manuscript itself what each of the criteria in "STAPH" stands for rather than stating only in the abstract and tables.

Response:

Thank you for your excellent suggestion. To help readers understand the STAPH score earlier, we have revised the last paragraph of the INTRODUCTION (see lines 83–88), in addition to mentioning it in the abstract and tables. Additionally, we have added the criteria of the STAPH score to the METHODS section of the main text (see lines 125–135).

Reviewer #2-minor comment 2

Line 105 states that author TH has 9 years of experience in Gram staining. Please also state whether this experience includes microscopic interpretation.

Response:

Thank you for your comment. We appreciate the opportunity to clarify the author's experience. Author TH's 9 years of experience in Gram staining does indeed include routine microscopic interpretation of blood culture specimens. This detail has been added to the manuscript as follows:

“Examiner 1 (TH) is a physician in the Department of Infectious Diseases with 9 years of experience in Gram staining, including routine microscopic interpretation of blood culture specimens.”

Reviewer #2-minor comment 3

Were any repeat patients included in the study? If so, please identify the number or state that no repeat patients were included.

Response: Thank you for your comment. The 148 samples analyzed in the study were collected from 68 patients. One patient had a second sample collected 37 days after the initial positive result, which grew a different organism. This detail has been revised in the manuscript's RESULTS section as

follows:

“A total of 148 specimens were analyzed, collected from 68 patients. Of these, 49 were identified as SA (33.1%) and 99 as CoNS (66.9%). One patient was a repeat case, with a second sample collected 37 days after the initial positive result, which grew a different organism.”

Re: Spectrum01223-24R1 (Rapid differentiation of *Staphylococcus aureus* in blood cultures using the STAPH score: a prospective observational study)

Dear Dr. Toshimasa Hayashi:

Your manuscript has been accepted, and I am forwarding it to the ASM production staff for publication. Your paper will first be checked to make sure all elements meet the technical requirements. ASM staff will contact you if anything needs to be revised before copyediting and production can begin. Otherwise, you will be notified when your proofs are ready to be viewed.

Please address final comment from Reviewer: Add "Examiner 1 (TH) is a physician in the Department of Infectious Diseases with 9 years of experience in Gram staining, including routine microscopic interpretation of blood culture specimens." to line 118 as described in the response to reviewer comments.

Sincerely,
Deena Altman
Editor
Microbiology Spectrum

Reviewer #1 (Comments for the Author):

The authors have addressed the question I previously raised. The study presents a standardized and innovative scoring system for SA assessment, which, although requiring further clinical validation to confirm its validity and practicality, is currently characterized by simplicity and practicality in terms of primary assessment.

Reviewer #2 (Comments for the Author):

The authors present a revised manuscript detailing the use of the STAPH score system to differentiate *Staphylococcus aureus* from coag-negative staph in blood cultures

Comment 1. Please add "Examiner 1 (TH) is a physician in the Department of Infectious Diseases with 9 years of experience in Gram staining, including routine microscopic interpretation of blood culture specimens." to line 118 as described in the response to reviewer comments.